# Normative reference values of the handgrip strength for the Portuguese workers

**Sarah Moreira Fernandes Bernardes**[1]ʘ*, **Ana Assunção**[1], **Carlos Fujão**[2,3],
**Filomena Carnide**[1]ʘ

1 Biomechanics and Functional Morphology Laboratory, CIPER, Sports & Health Department, Faculty of Human Kinetics, University of Lisbon, Lisbon, Portugal, 2 Volkswagen Autoeuropa–Area of Industrial Engineering and Lean Management, Palmela, Portugal, 3 Institute of Education and Science, Universitas, Lisbon, Lisbon, Portugal

ʘ These authors contributed equally to this work.
* sarah.mfber@gmail.com

**Data Availability Statement:** Additional data files are available from the figshare database: https://doi.org/10.6084/m9.figshare.12333293.v1.

**Funding:** This work was partly supported by Foundation for National Scientific Computation,

## Abstract

### Introduction

The active workforce is increasingly aging. However workload, as well as working time and intensity, sometimes remains unchanged. This can be an even more critical situation in older people, since occupational exposure associated with aging, will further reduce the muscle's ability to generate energy, which in turn facilitates the development of these age-related syndromes. This study aims to identify the normative values of handgrip strength for Portuguese workers in the automotive industry.

### Methods

About 1225 employees were invited to participate in the study. The final sample consisted of 656 employees in the assembly area. The handgrip strength was measured in kilograms (kg) using the Jamar digital dynamometer. Sex-specific profiles of handgrip strength were designed by the Ordinary Least Square regression (OLS) analysis, where height, age, age squared, and height squared are entered into the models as determining factors of the maximum grip strength in both female and male groups.

### Results

The peak mean values of handgrip strength in the group of women was 34 kg in the age group of 35–39 years, and in the group of men the peak mean was 52 kg in the age group of 25–34 years. The most pronounced decline in the female group appears in the age of 30–34 years of about 4 kg and the male group the decline occurs at 2kg below the peak force, in the age group of 40–57 year-olds. This study used a cut-off at 2 SD below by the sex-specific peak mean.

### Conclusion

Normative values can help delineate the career path of workers because they portray risk values according to age, height, and gender. The normative values assist health and

under Grant UIDB / 00447/2020 to CIPER Interdisciplinary Center for the Study of Human Performance (unit 447), Ph.D. grant by Coordination of Superior Level Staff Improvement (CAPES, Brazil) 011990 / 2013-09. The funders had no role in study design, data collection and analysis, decision to publish or preparation of the manuscript.The funder (Volkswagen Autoeuropa) provided support in the form of authors' salaries and / or research materials for authors (Carlos Fujão), but did not have any additional role in the study design, data collection and analysis, decision to publish or preparation of the manuscript. The specific roles of these authors are articulated in the 'author contributions' section.

**Competing interests:** Author (Carlos Fujão) were employed by Volkswagen Autoeuropa which supervision the project in Volkswagen Autoeuropa. Volkswagen Autoeuropa did not have any additional role in the study design, data collection and analysis, decision to publish or preparation of the manuscript. The other authors have no competing interest conflict. This does not alter our adherence to PLOS ONE policies on sharing data and materials.

engineering professionals and ergonomists in adjusting task demands to the morphological and strength characteristics of the workers.

## Introduction

Aging occurs at an exponential growth rate worldwide, characterized by the process of progressive decline in cellular, physical, and mental capacities [1]. This process is closely related to the incidence of diseases, such as sarcopenia (progressive and significant loss of muscle mass and strength) and frailty syndrome (age-related decline in the physiological system, affecting strength and resistance, increasing the risk of falls, dependence or death) [1–4].

These syndromes can appear early on and be aggravated by the exposure to the demands of work during the active course of professional life [2–7]. Furthermore, this has become more critical since the active workforce is increasingly aging. In contrast, workloads, as well as time and intensity of work, sometimes remain unchanged, in the older and in the middle-aged population [1–7].

As a consequence of unchanged occupational exposure, in middle-aged and older workers, the muscle's ability to generate energy tends to be reduced which, in turn, facilitates the early development of these age-related syndromes and as so-called work-related musculoskeletal disorders [1,8,9].

One of the key measures used for the diagnosis of these age-related syndromes is the handgrip strength test (HGS), which is measured by the static force exerted by the hand when holding and tightening a dynamometer [10]. The HGS is considered a biomarker of healthy aging [11] and an indicator of general muscle strength [12]. Also, it has excellent inter-rater reliability, is easy to apply, and has a low cost [4,13–15].

Studies are rarely carried out on active-age workers who are exposed to highly demanding activities, such as in the automotive industry.

Reference values regarding handgrip strength in the automotive industry are few and feature small samples, being n 161, in the Australian population and 206 individuals in the British population [16,17]. By contrast there are studies with larger sampling, such as the study carried out in Germany with a sample of 11,790 people aged 17–90 years, but the type of occupational activity that people were allocated to was not categorized. This Germany study validated the strong association between body height and increased handgrip strength, with each 10 cm in height associated with an increase in handgrip strength of 2 to 4 kg [18]. In another study carried out in the British population, the sample was 1645 people but did not report the activities the participants performed.

A study developed in the American population which is widely used in research because it was one of the first to provide normative values of handgrip strength, had a sample of 638 people between 20 and 90 years of age. However this study did not categorize the type of activities and occupational factors the participants were exposed to [19].

Therefore, studying the HGS measure in active-age workers can be fundamental in avoiding the early appearance of age-related syndromes and even the early appearance of work-related musculoskeletal disorders, which would influence the decrease in premature retirements.

Thus, this study aims to identify the normative values of handgrip strength for Portuguese workers in the automotive industry.

## Materials and methods

The scientific committee of the Faculty of Human Kinetics from the University of Lisbon has approved the study protocol (protocol number 30/2019). All workers were informed about the purpose and procedures of the study and given their written informed consent.

## Sample

A cross-sectional study was conducted in the area of assembly within the automotive industry. In the present study, the sample was recruited from a population of 1225 direct workers, using the randomization criterion, from the entire list of employees in the assembly area, provided by the industry's occupational area. The entire study took place at assembly facilities, lasting 20 weeks from September 2018 to January 2019. For a transversal study considering 5% of alpha error, 80% of power, and an effect size of 0.7, the sample must include 720 participants [20].

In this present study for the sample size calculation, participants were stratified by gender and age. Female workers were divided into five age groups and male workers into six age groups. Height levels were restricted to 146–177 cm for women and 150–190 cm for men.

To ensure normative values for healthy workers, the following exclusion criteria were applied: a) minimum value of 10 kg of handgrip force (exclusion workers; b) presentation of any medical restriction or occupational disease from the industry's occupational department; c) SF-12 criteria (Short Form Health Survey) with a score lower than 5% of the physical component score of the quality of life scale [18].

## Measurements

**Questionnaire SF-12 (Short Form Health Survey).** The SF-12 is composed of 12 items organized according to a Likert scale and includes physical components score (PCS) and mental components scores (MCS). The physical dimension comprises items related to physical function, physical performance, pain, and health in general, and the mental dimension covers mental health, emotional performance, social function, and vitality [18,21]. SF 12 is considered a measure of high reliability concerning physical and mental aspects, in a study for HGS values with a large sample, and this instrument was used as an exclusion criterion [18].

**Anthropometry.** For the height measurement, an upright position was obtained, participant-centred position on the tape, in relation to the stadiometer, footwear, arms extended along the body, feet joined or slightly separated. The head was oriented according to the Frankfurt plane, parallel to the ground, regardless of the worker's posture [22]. Due to the European safety norms industry setting, to height measure were removed 3 centimetres to compensate the height of the work shoes sole.

**Handgrip strength (HGS).** For the manual handgrip strength test, the standard position (the worker was in the sitting position, with the arms in abduction, forearms in pronation, and the hand in neutral position, elbow flexed at 90° degrees) was used for all participants [18,19,23]. The grip strength was measured in kilograms (kg) using the Jamar digital dynamometer [24]. Two measurements were performed in both hands. Position 2 of the Jamar dynamometer was used, because is the most appropriate position to measure the handgrip strength [19,24]. The maximum value obtained with either hand is used as a summary measure of a person's isometric strength of the hand and forearm muscles [25,26]. The Jamar dynamometer is extremely used, which is validated as a gold standard, with high test-retest reproducibility (r> 0.8) and excellent reliability (r = 0.98) [27].

## Data analyses

In order to perform the final sample, the eligibility criterion was applied. This was based on the SF-12 PCS score, standardized from the z-standardized with an average value of 36 and SD 2.35, defining the criterion for 5% below the PCS average.

Descriptive statistical analysis was used to determine mean (M), standard deviation (SD) and median values (ME) of the handgrip strength, stratified by five age groups of the women and the six age groups of men. Sex-specific profiles of handgrip strength were designed by the

Ordinary Least Square regression (OLS) analysis [28], where height, age, age squared and height squared (independent variables) are entered in the models as determinant factors of the maximum grip strength (dependent variables) in both female and male groups. The objective of this statistical technique was to verify the mean peak values for women and men groups. A p-value of 0.05 was set as a statistically relevant result.

Afterwards, the cut-off values were calculated in order to identify workers with weak grip strength. These values were defined as 1SD and 2SD below the mean peak value and stratified by gender [9,29]. The resulting values were plotted from the age groups defined by height.

Finally, handgrip strength measurements were first standardized for age and body height. The risk threshold was determined by subtracting 1 SD from the mean value of height-adjusted handgrip strength and age group [9]. The standardized measures of handgrip strength values were the z-standardized residuals (derived from sex-specific OLS regressions of hand-grip strength values in kg on age (in years) and body height (in cm).

## Results

The sample was composed by 634 workers who had accepted to participate in the study. From those, 39 workers were excluded because they didn't fulfil the eligibility criteria, namely the 5% the physical component score (PCS) based in SF-12. This criterion was chosen to restrict the test population to healthy employees only. In addition to the SF-12 standards, it was also verified if the list of employees had any medical restriction from the occupational health department of the automotive industry.

So, the final sample included 617 workers, mainly men (~74%), with a mean age of 33±8.58 years and a mean height of 173±6.50 cm; women presented a mean age of 32±8.03 years and a mean height of 160±5.95 cm.

In the present study, the only anthropometric measure we covered was height. The average height of women, according to the age groups, was 20–24 and 35–39 (1.61 cm), 25–29 and 30–34 years (1.62 cm), and in the 40–55 group, the mean height was 1.59 cm. The average height by the age groups of men was 20–24 (1.75 cm), 25–59, 30–34, and 35–39 (1.73 cm) and 40–44 and 45–57 (1.71 cm) (S1 Table). Table 1 shows the descriptive analysis of the age, heights, right handgrip measure 1, left handgrip measure 1, right handgrip measure 2, left handgrip measure 2 and the maximum measure handgrip, for all sample groups of men and women.

The mean peak of the handgrip strength was obtained from the OLS regression. The mean peak value of the handgrip strength for women was 34 kg and was reached in the age group of

**Table 1. Descriptive analysis of the all sample and the group of men and women.**

|  | All Sample (n 617) | | | | Male (n 458) | | | | Women (n 159) | | | |
|---|---|---|---|---|---|---|---|---|---|---|---|---|
|  | Min | Max | x | SD | Min | Max | x | SD | Min | Max | x | SD |
| Age (years) | 20.0 | 57.0 | 32.7 | 8.45 | 20.0 | 57.0 | 33.1 | 8.58 | 20.0 | 55.0 | 31.9 | 8.03 |
| Height (cm) | 146 | 193 | 170 | 8.45 | 154 | 193 | 173 | 6.50 | 146 | 179 | 160 | 5.85 |
| R Hand 1 (kg) | 11.3 | 82.7 | 42.3 | 12.2 | 16.0 | 82.7 | 46.9 | 10.1 | 11.3 | 50.6 | 29.1 | 6.75 |
| L Hand 1 (kg) | 10.9 | 80.1 | 40.3 | 11.4 | 19.9 | 80.1 | 44.7 | 9.43 | 10.9 | 49.9 | 27.9 | 6.55 |
| R Hand 2 (kg) | 11.6 | 79.7 | 43.2 | 12.0 | 18.1 | 79.7 | 47.8 | 9.80 | 11.6 | 50.8 | 29.8 | 6.96 |
| L Hand 2 (kg) | 12.8 | 79.0 | 39.6 | 11.0 | 16.8 | 79.0 | 43.7 | 9.29 | 12.8 | 44.6 | 27.8 | 6.15 |
| Max. M (kg) | 17.7 | 82.5 | 45.5 | 11.9 | 20.1 | 82.5 | 50.4 | 9.30 | 17.7 | 50.8 | 31.4 | 6.36 |

Max. M = maximum measure of HGS; R Hand 1 and 2 = Right Hand measure of HGS 1 and 2; L Hand 1 and 2 = Left Hand HGS measure of HGS 1 and 2. As stated in the grip strength protocol, there were two measurements with the right hand and the left hand (in totally four-measure). However, only the maximum measure that was considered for the study.

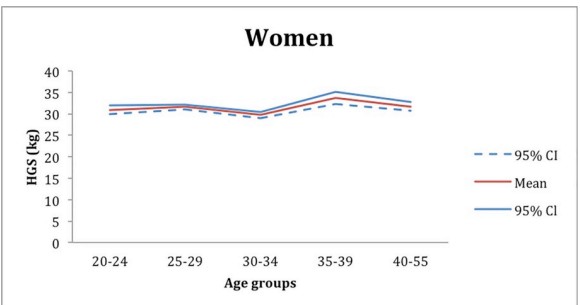

**Fig 1. Mean peak values of handgrip strength extracted from the OLS regression in the women group workers.**

35–39 years (Fig 1) in contrast, for men, the mean peak value of handgrip strength was of about 52 kg in the age group of 25–34 years (Fig 2).

The most pronounced HGF peak decrease appeared in women in the age group of 30–34 year-olds, with 4kg (Fig 1) below the mean peak strength. For the men, this decline was 2kg, in the age group between the 40–57 years (Fig 2).

Furthermore, the prevalence of workers in each of the weak handgrip strength groups was determined through 1 SD and 2 SD below the peak mean values, also in relation to the low grip strength cut-off for men at 27 kg and for women at 16 kg, based on Cruz-Jentoft et al [9]. The age group of 20–24 year-olds in the women's group had the values of 1 SD and 2 SD, respectively, 25.4 kg and 20.1 kg. The female age group of 30–34 year-olds i, they had the lowest values of deviations, and in the 2 SD (15.1 kg), they were below the cut-off (Fig 3).

The age group of 35–39 year-olds in the men's group had the values of 1 SD and 2 SD, respectively, 42.7 kg and 34.6 kg. The lowest values of 1 SD and 2 SD found in the group of men were in the 45–57 age group with 38.8 kg and 29.9 kg, 1 and 2 SD respectively. However, no age groups in the men's group are below the cut-off line (Fig 4).

The normative values for Portuguese workers are presented in Tables 2 and 3, separately for women and men. The female workers were divided into 5 age groups (20–24, 25–29, 30–34, 35–39, 40–55) and in the male workers were distributed by 6 age groups (20–24, 25–29, 30–34, 35–39, 40–44, 45–57). In the group of women, we chose to divide it into five age groups, because the population over 40 was not in large number.

The maximum value of the handgrip strength in the female groups and in the age group of 35–39 years, with a height greater than (+) 171 cm, was 39.5 kg. Moreover, the minimum value appeared in the 40–55 age group with a height between 146-150cm, and 25.1kg (Table 2).

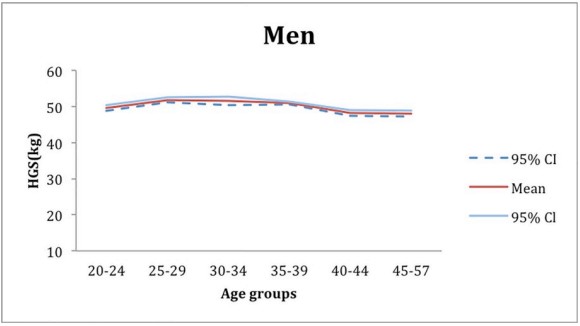

**Fig 2. Mean peak values of handgrip strength extracted from the OLS regression in the male group workers.**

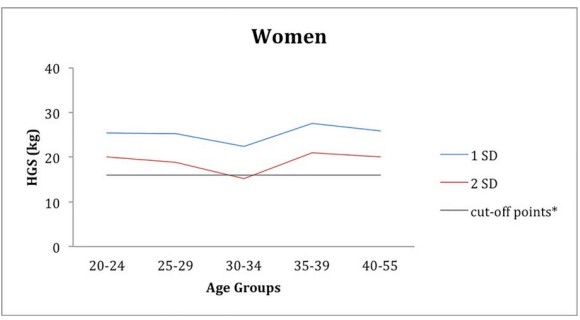

**Fig 3. Distribution of female workers according to the mean grip strength values (below 1 SD and 2 SD), and the relation to the cut-off, according to the age groups.**

The maximum value of the handgrip strength was 63.0 kg for the male workers in the age group of 20–24 years and height higher than (+) 185 cm. The minimum value was about 40.5 kg in the age group of 45–57 years at the height of 154-159cm (Table 3).

Table 4 presents the proportion of workers, according to age and height groups, of both genders, stratified by standard deviations, from the Z-standardized values of the handgrip strength, obtained through the Ordinary least square regression models.

It was observed that the largest number of workers were in the reference group, and, in the group of men, only 4.7% of the sample of the workers were between 1.5 - <3.0 SD, below the mean values. For the women only 6.4% were between 1.5 - <3.0 SD below the mean values for gender (Table 4).

## Discussion

This study aims to establish normative values for the handgrip strength of Portuguese workers in the automotive industry in order to identify low handgrip strength thresholds. Such limits can assist in identifying, declining aspects of muscle capacity, and even form an alert to check for the possible development of frailty and sarcopenia syndrome. These declines in muscle capacity and the warning for syndromes related to aging are likely since the grip strength measure can be used as a predictor of whole-body strength [30]. As for frailty and sarcopenia, the HGS is one of the parameters to identify them [9].

The limit values were reported and stratified by sex, age, and body height. Stratification by height is essential as it has an influential role in handgrip strength. There are reports in studies that every 10 cm body height can lead to a 2–4 kg increase in handgrip strength [18]. However, in the present study, when comparing the lowest and highest height, in both gender groups

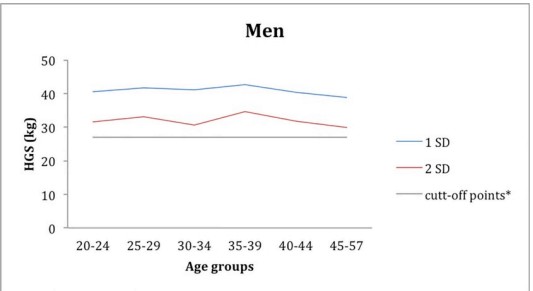

**Fig 4. Distribution of male workers according to the mean grip strength values (below 1 SD and 2 SD), and the relation to the cut-off, according to the age groups.**

**Table 2. Normative reference values of handgrip strength for women workers.**

| Age (years) | Height (cm) | HGM (Kg) | SD | Risk Threshold |
|---|---|---|---|---|
| 20–24* n (31) | 151–155 | 31.0 | 6.00 | 25.0 |
| | 156–160 | 29.5 | 4.84 | 24.7 |
| | 161–165 | 31.2 | 3.53 | 27.7 |
| | 166–170 | 29.9 | 7.06 | 22.8 |
| | +171 | 36.3 | 2.61 | 33.7 |
| 25–29 n (51) | 146–150 | 27.2 | 5.30 | 21.9 |
| | 151–155 | 30.1 | 3.19 | 26.9 |
| | 156–160 | 31.0 | 5.96 | 25.0 |
| | 161–165 | 30.8 | 6.41 | 24.4 |
| | 166–170 | 35.7 | 8.52 | 27.2 |
| | +171 | 33.2 | 12.8 | 20.4 |
| 30–34* n (30) | 151–155 | 27.2 | 2.28 | 24.9 |
| | 156–160 | 28.2 | 4.71 | 23.5 |
| | 161–165 | 30.6 | 9.35 | 21.3 |
| | 166–170 | 31.2 | 4.91 | 26.3 |
| | +171 | 30.1 | 11.4 | 18.7 |
| 35–39* n (22) | 146–150 | 29.2 | 0.21 | 29.0 |
| | 151–155 | 38.2 | 0.56 | 37.6 |
| | 161–165 | 32.9 | 2.15 | 30.8 |
| | 166–170 | 36.2 | 14.0 | 22.2 |
| | +171 | 39.5 | 4.59 | 34.9 |
| 40–55* n (34) | 146–150 | 25.1 | 3.66 | 21.4 |
| | 151–155 | 32.1 | 8.47 | 23.6 |
| | 156–160 | 32.5 | 5.47 | 27.0 |
| | 161–165 | 32.1 | 5.60 | 26.5 |
| | 166–170 | 35.6 | 1.62 | 34.0 |

At the intersection between the height of 146–150 cm and age between 20–24 and 30–34 years-old, the height of 156–155 cm and age between 35–39 years-old, and the height of +171 cm and age between 40–55 years-old, no participant was found such characteristic, and therefore, this height was withdrawn from this age group.

(Tables 1 and 2), it is interesting to note that sometimes the difference is greater than 2 to 4 kg of strength [18]. This may also be related to the type of activity that workers perform in the assembly area [31], where the muscle groups of the forearm region, may exert more effort [1]. It is important to note that the study population is limited to healthy workers.

In the automotive industry, or in other occupational settings, the use of height measurement can be an essential factor when designing work conditions tailored to older workers. For example, in activities that require gripping of the objects above the head level, it would be appropriate to change the layout or work plan height in order to allow senior workers to perform them without constrains. Such preventive measure could prevent the emergence of musculoskeletal disorders and even the early onset of sarcopenia due to micro trauma developed by the impact of task demands [16,32–36].

Another interesting finding in the present study has found in the group of women between 30 and 34 years old, where there was a 4 kg drop in handgrip strength when compared to women in the age group of 35 to 39 years. This situation can be explained by the seniority in the company, where the age group that comprised the highest average peak force (35–39 years)

**Table 3. Normative reference values of handgrip strength for men workers.**

| Age (years) | Height (cm) | HGM (Kg) | SD | Risk Threshold |
|---|---|---|---|---|
| 20–24* n(80) | 160–164 | 47.6 | 4.10 | 43.5 |
| | 165–169 | 46.1 | 8.29 | 37.8 |
| | 170–174 | 48.3 | 9.76 | 38.5 |
| | 175–179 | 50.5 | 8.61 | 41.9 |
| | 180–184 | 52.4 | 7.14 | 45.3 |
| | 185 | 63.0 | 6.61 | 56.4 |
| 25–29 n (102) | 154–159 | 46.8 | 6.15 | 40.7 |
| | 160–164 | 47.8 | 5.41 | 42.4 |
| | 165–169 | 47.0 | 5.83 | 41.2 |
| | 170–174 | 51.7 | 9.20 | 42.5 |
| | 175–179 | 52.6 | 10.1 | 42.5 |
| | 180–184 | 56.7 | 8.53 | 48.2 |
| | 185 | 55.7 | 9.43 | 46.3 |
| 30–34* n (80) | 160–164 | 43.8 | 1.60 | 42.2 |
| | 165–169 | 46.2 | 10.7 | 35.5 |
| | 170–174 | 48.6 | 8.63 | 40.0 |
| | 175–179 | 54.2 | 9.55 | 44.7 |
| | 180–184 | 61.2 | 10.4 | 50.8 |
| | 185 | 62.1 | 8.44 | 53.7 |
| 35–39* n (73) | 160–164 | 43.9 | 13.2 | 30.7 |
| | 165–169 | 52.5 | 11.1 | 41.4 |
| | 170–174 | 48.6 | 8.61 | 40.0 |
| | 175–179 | 49.1 | 7.70 | 41.4 |
| | 180–184 | 55.9 | 5.68 | 50.2 |
| | 185 | 55.1 | 11.4 | 43.7 |
| 40–44 n (54) | 154–159 | 46.0 | 3.15 | 42.9 |
| | 160–164 | 44.3 | 9.32 | 35.0 |
| | 165–169 | 46.8 | 9.92 | 36.9 |
| | 170–174 | 51.0 | 7.93 | 43.1 |
| | 175–179 | 51.8 | 8.29 | 43.5 |
| | 180–184 | 53.8 | 4.16 | 49.6 |
| | 185 | 44.2 | 3.46 | 40.7 |
| 45–57 n (60) | 154–159 | 40.5 | 10.5 | 30.0 |
| | 160–164 | 49.9 | 10.7 | 39.2 |
| | 165–169 | 44.6 | 5.03 | 39.6 |
| | 170–174 | 46.8 | 11.2 | 35.6 |
| | 175–179 | 52.7 | 9.27 | 43.4 |
| | 180–184 | 50.4 | 8.85 | 41.6 |
| | 185 | 57.6 | 6.78 | 50.8 |

At the intersection between the height of 154–159 cm and the age between 20–24; 30–34,and 35–39 years old, no participant was found with such characteristic, and therefore, this height was removed from this age group.

was in the company for only two (2) years and was therefore less impacted by work demands, which remain the same for young and older workers [16,31,32,37,38]. In the male group, the decline is in the age group of 40–57 years, which was expected, since it is in this age group that the most characteristic decline occurs when compared to the other age groups [10,12,23,39,40].

**Table 4. Distribution of workers by gender and age groups according the height–z-standardized handgrip strength.**

|  | n Male | % | n Female | % |
|---|---|---|---|---|
| Reference group sM(+/0.5SD) | 182 | 40.9 | 67 | 42.7 |
| (1) 0.5SD <1.0 SD below sM | 68 | 15.3 | 28 | 17.8 |
| (2) 1.0 SD <1.5 SD below sM | 42 | 9.40 | 9 | 5.70 |
| (3) 1.5 SD <3.0 SD below sM | 21 | 4.70 | 10 | 6.40 |
| (4) 0.5 SD <1.0 SD above sM | 65 | 14.6 | 15 | 9.60 |
| (5) 1.0 SD <1.5 SD above sM | 40 | 9.00 | 17 | 10.8 |
| (6) 1.5 SD <3.0 SD above sM | 27 | 6.10 | 11 | 7.00 |
| **Total** | 445 | 91 | 157 | 100 |

The sample consisted of men and women between the ages of 20 and 57 years. In the men's group, the body height has restricted in 160–200 cm. And the women group the body height is restricted from 146 to 181 cm. And the HGS measured is limited between 10 and 75 kg. The standardized handgrip strength was obtained from the Ordinary least square and using the z-standardized residuals (M = 0 SD = 0.96).

To our knowledge, our study is the first to produce normative handgrip strength values for the Portuguese workers in automotive industry, so we chose to compare the results of our study with 4 other previously published international studies. We considered the differences in mean peak values from previous studies with our average grip strength values expressed as a percentage of our value and / or in kg.

In the first previously published study conducted in the German population, between the ages of 17 and 90 years, in both sexes; compared to the present study, the group of German men has mean peak handgrip strength of 2% higher than the male population of the present study. When compared to the present study, Portuguese women are 0.5% below the average peak force in comparison to the German women group [18].

The second study conducted in the English population, where 12 major reviews were compiled for the creation of the normative values of handgrip strength. The values of the peak handgrip strength were different when compared to the present study. Both the English men and women groups are 3% below the mean peak strength value observed in the Portuguese workers [23].

In a study conducted in the American population, normative values of handgrip strength were determined in the seven age groups. The handgrip strength peak of the American male population was 3.2% higher than the present male population. In the women's group, when compared to the current study group, the average grip strength was 1.7% lower in all age groups [19].

By comparing the statistics from America, English, and German, with the population of the present study, we can infer that small differences in strength could be closely related to the type of activity (work demands) developed in the automotive sector. Effectively, there are several high demanding tasks, characterized by handling loads, unfavorable static postures, repetitive upper limb movements of force, influence on the occurrence of micro trauma surgery in the wrist, hand and elbow regions, interfering with the handgrip strength [1,31–33,41–43].

In another study of 187 male workers in the English automotive industry, the average peak handgrip strength was found in the age group of 40–44 years with 47kg grip strength [17]. Thus, the male population of the present study has an average peak handgrip strength that is 6 kg higher than the English car industry population. We can infer from the report of the author of the fourth study, that the situation of lower grip strength may be associated with the lack of calibration of the dynamometers used [17,44].

Our study has shown that grip strength has a peak increase in early adulthood (20–24 years) in both genders, and then enters a less-pronounced decline with advancing age (from the age of 40). However, as explained above, the group of women (30–34 years) due to occupational exposure had a more pronounced decline [16].

Also this study found a high prevalence of weak grip strength based on 1 and 2 SD and cut-off points based on [9]. In the 30–34 age group for women and in the 45–57 age group with 15.2 kg for women and 29.9 kg for men, thus producing more discriminatory cutoff values for grip strength by 40% for women (30–34 years) and 23.2% in the group of men aged 45–57 years (Figs 3 and 4). The cut-off could be a significant ally for the automotive industry, where these age groups in both genders should be monitored more frequently to prevent the onset of sarcopenia and frailty syndrome [9].

## Conclusion

Normative values can help delineate the career path of workers because they represent risk values according to age, height, and gender. These thresholds can be very useful to help on the adjustment of work conditions to the morphological and strength characteristics of the worker. Thus, it is possible to design or redesign the conditions work processes associated with the predictive values of HGS and the implementation of the workers' clinical surveillance system through periodic using the HGS test.

## Supporting information

**S1 Table. Table 1 descriptive analysis of height for age groups in women and men.** (DOCX)

**S1 Data.** (XLSX)

## Acknowledgments

The authors are grateful to all the workers who voluntarily participated in this study.

## Author Contributions

**Conceptualization:** Sarah Moreira Fernandes Bernardes, Filomena Carnide.

**Data curation:** Sarah Moreira Fernandes Bernardes, Ana Assunção.

**Formal analysis:** Sarah Moreira Fernandes Bernardes, Ana Assunção, Filomena Carnide.

**Investigation:** Sarah Moreira Fernandes Bernardes.

**Methodology:** Sarah Moreira Fernandes Bernardes, Filomena Carnide.

**Project administration:** Sarah Moreira Fernandes Bernardes.

**Resources:** Sarah Moreira Fernandes Bernardes.

**Software:** Sarah Moreira Fernandes Bernardes.

**Supervision:** Carlos Fujão, Filomena Carnide.

**Validation:** Sarah Moreira Fernandes Bernardes, Filomena Carnide.

**Visualization:** Sarah Moreira Fernandes Bernardes, Filomena Carnide.

**Writing – original draft:** Sarah Moreira Fernandes Bernardes, Filomena Carnide.

**Writing – review & editing:** Sarah Moreira Fernandes Bernardes, Filomena Carnide.

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
