## [Decision Letter · Decision Letter 0]

14 May 2020

PONE-D-20-06221

Normative reference values of the handgrip strength for the Portuguese workers.

PLOS ONE

Dear Dr Bernardes

Thank you for submitting your manuscript to PLOS ONE. After careful consideration, we feel that it has merit but does not fully meet PLOS ONE’s publication criteria as it currently stands. Therefore, we invite you to submit a revised version of the manuscript that addresses the points raised during the review process.

According to the opinion of the reviewers, the manuscript brings an interesting subject, however, it needs to be reviewed on several topics. After all these changes, the authors can resubmit the manuscript for a new evaluation.

We would appreciate receiving your revised manuscript by may-23. To enhance the reproducibility of your results, we recommend that if applicable you deposit your laboratory protocols in protocols.io, where a protocol can be assigned its own identifier (DOI) such that it can be cited independently in the future. For instructions see: http://journals.plos.org/plosone/s/submission-guidelines#loc-laboratory-protocols

We look forward to receiving your revised manuscript.

Kind regards,

Anderson Saranz Zago, PhD

Academic Editor

PLOS ONE

2. In your Methods section, please provide additional information about the participant recruitment method and the demographic details of your participants.

Please ensure you have provided sufficient details to replicate the analyses such as:

a) the recruitment date range (month and year),

b) a description of any inclusion/exclusion criteria that were applied to participant recruitment,

c) a table of relevant demographic details,

d) a statement as to whether your sample can be considered representative of a larger population,

e) a description of how participants were recruited, and

f) descriptions of where participants were recruited and where the research took place.

'NO authors have competing interests'

We note that one or more of the authors are employed by a commercial company: Volkswagen Autoeuropa.

5. Please ensure that you refer to Figure 4 in your text as, if accepted, production will need this reference to link the reader to the figure.

6. Please include captions for your Supporting Information files at the end of your manuscript, and update any in-text citations to match accordingly. Please see our Supporting Information guidelines for more information: http://journals.plos.org/plosone/s/supporting-information

**Comments to the Author**

1. Is the manuscript technically sound, and do the data support the conclusions?

Reviewer #1: Yes

Reviewer #2: No

2. Has the statistical analysis been performed appropriately and rigorously? 

Reviewer #1: Yes

Reviewer #2: Yes

3. Have the authors made all data underlying the findings in their manuscript fully available?

Reviewer #1: Yes

Reviewer #2: Yes

4. Is the manuscript presented in an intelligible fashion and written in standard English?

Reviewer #1: Yes

Reviewer #2: No

5. Review Comments to the Author

Reviewer #1: Dear Authors,

The research aimed to identify the normative values of handgrip strength for Portuguese workers in the automotive industry. The research has the potential to improve working conditions, prevent occupational disease, premature retirements and, consequently improve the quality of live for Portuguese workers.

Although the research is well written and the message is clear, minor adjustments are needed, which will be presented below:

Abstract: do not repeat the words that were used in the title as key terms.

Line 161: correct reference 9 (remove parentheses and inform author).

Results: The description of the sample was well done, but I missed the number of individuals per age group. I request that the authors add this information.

The results were well explored as text, however, the tables and figures can be improved.

Tables and figures:

Complement in the title the characteristics of the studied population: Portuguese women and men workers in the automotive industry.

Include the description of the abbreviation OR avoid abbreviation and include the complete nomenclature.

Include the number of people per group (“N”), when necessary.

Figures 3a and 3b: The figures are not clear, I suggest inserting the units of measurement of the scales.

The manuscript needs minor modifications before being accepted for publication. I hope that my suggestions contribute to improving the manuscript.

Sincerely.

Reviewer #2: The present study aimed to identify the normative values of handgrip strength for Portuguese workers in the automotive industry.

Abstract

Results: Please, verify age range for women regarding force declining. The abstract indicated force starting to declines for ages bellow the range that force reaches the peak … is it correct?

Conclusion: authors re-introduced the issue and suggested applications in very broad sense. Please, re-write the text according to the evidences of your results and make specific appointments concerning occupational care of Portuguese workers.

Introduction

The present study aimed “to identify the normative values of handgrip strength for Portuguese workers”. According to the authors, the assessment of strength level would support the diagnosis of muscle syndromes related to aging process (e.g.: frailty and sarcopenia), and handgrip scores is reliable to parametrize the development of syndromes affecting muscle ability to generate force accordingly. Moreover, authors included some paragraphs (apart of general introduction, which does not match PLOS One guidelines) to specify how different countries developed their own reference for handgrip strength.

However, author did not give enough information to support the theoretical link between handgrip scores of muscle force and muscle weakness/disability to perform a given occupational function. Furthermore, author fails to state the need to develop an index of force for each country, and the reasons for the lack of confidence in worldwide scores.

Methods

pp. 56 – 57: why to elect only workers from final assembly area?

pp. 60 – 65: text discussing the sample size. It is not appropriate in method section.

pp. 67-69: include the age range, numbers of workers, average weight and height for each age group.

pp. 70-74: include the numbers of participants excluded according each criteria.

p. 76: Methods (again?) …. Measurements (?)

pp. 114 – 117: identify the dependent parameter.

p. 115: OLS require alpha number to determine include and excluded variables.

pp. 120-122: support the choice of 1SD and 2SD to define cut-off values.

pp. 124 – 125: support the choice of 1SD to define risk threshold.

Results

pp. 131 – 140: describe the process of sample selection and should be located accordingly. Decimal cases should be separated by points.

pp. 141-144: I can’t observe differences in force values when compared the age groups for both female (Fig 1) and male (Fig 2). Did authors proceeded statistical test to compare force between groups?

All Figures should be revised. The units are not present, as well as, “Y” title.

p. 150: I am not sure that the term “decline” fits well to describe force profile variations between age groups. If authors considered such score the worsts, then force has not age as independent factor among females.

p. 163: revise kilograms symbol.

Figures 3a and 3b aren’t parts of the same figure.

Discussion

pp. 231-233: the findings are limited to the establishment of reference values. There is no results to confirm that workers having force values bellow the risk threshold presented sarcopenia or frailty syndrome. Please, revised this statement or support it better from your results.

pp. 236-237: This statement could not be corroborated from the results of Tables 1 and 2. Please, discuss it better.

p. 272: “malepopulation” means “male population”?

pp. 289 – 300: results for peak of force, force decrement, and cut-off points were just comment and not compared to others results from literature.

In discussion section, authors limited to explain differences regarding the references of force for handgrip test by comparing with others similar studies, or even reinforce main results. The applications to parametrize muscle syndromes and support occupational adjustments were discussed superficially.

Conclusion

pp. 304-309: author concluded about possible applications of the results. Most of these applications had no results to support the appointments. For example: (1) the ability to perform work above head was not related to handgrip force, and (2) the clinical relevance of handgrip force ability or disability to endorse worker relocation should be the proposition for future studies.

6. PLOS authors have the option to publish the peer review history of their article (what does this mean?). If published, this will include your full peer review and any attached files.

Reviewer #1: No

Reviewer #2: No

---

## [Author Response · Author response to Decision Letter 0]

23 May 2020

Dear Reviewers,

Thank you very much for the positive evaluation of our paper and valuable comments that helped us improve it. Many of these comments have embarrassed us, we should see these imperfections ourselves.

We have done our best to make the paper really good. We hope that it will meet your expectations. 

We have learned a lot thanks to your comments.

Thank you

1- Reviewer Editor

https://journals.plos.org/plosone/s/file?id=wjVg/PLOSOne_formatting_sample_main_body.pdf andhttps://journals.plos.org/plosone/s/file?id=ba62/PLOSOne_formatting_sample_title_authors_affiliations.pdf

 Answer: It has already been changed in the article and the system of Plos One.

2. In your Methods section, please provide additional information about the participant recruitment method and the demographic details of your participants.

Please ensure you have provided sufficient details to replicate the analyses such as:

a) the recruitment date range (month and year): 

Answer: It was added to the text: 

“The entire study took place at the assembly area facilities, lasting 20 weeks from September 2018 to January 2019”

b) a description of any inclusion/exclusion criteria that were applied to participant recruitment

Answer: We add detailed information about exclusion criteria:

“To ensure normative values for healthy workers, the following exclusion criteria were applied: a) minimum value of 10 kg of handgrip force; b) present any medical restriction and occupational disease from the industry's occupational department; c) SF-12 criteria (Short Form Health Survey) with a score lower than 5% of the physical component score of the quality of life scale (24).”

c) a table of relevant demographic details:

Answer: Due to the complexity of removing a worker from the assembly area, and as the study's objective was related to the handgrip measure, only the height measure was verified: We did not measure the weight. We have table 1, with descriptive analyzes of the total population, stratified by gender and describing age, the two measures of each hand (totaling four measures), and the maximum measure of handgrip strength (HGS).

Table 1. Descriptive analysis of the all sample and the group of men and women.

 All Sample (n 617) Men (n 458) Women (n 159)

 Min Max x SD Min Max x SD Min Max x SD

Age (years) 20.0 57.0 32.7 8.45 20.0 57.0 33.1 8.58 20.0 55.0 31.9 8.03

Height (cm) 146 193 170 8.45 154 193 173 6.50 146 179 160 5.85

Right Hand 1 (kg) 11.3 82.7 42.3 12.2 16.0 82.7 46.9 10.1 11.3 50.6 29.1 6.75

Left Hand 1 (kg) 10.9 80.1 40.3 11.4 19.9 80.1 44.7 9.43 10.9 49.9 27.9 6.55

Rigth Hand 2 (kg) 11.6 79.7 43.2 12.0 18.1 79.7 47.8 9.80 11.6 50.8 29.8 6.96

Left Hand 2 (kg) 12.8 79.0 39.6 11.0 16.8 79.0 43.7 9.29 12.8 44.6 27.8 6.15

Max. Measure (kg)* 17.7 82.5 45.5 11.9 20.1 82.5 50.4 9.30 17.7 50.8 31.4 6.36

Maximum measure of handgrip .As stated in the grip strength protocol, there were two measurements with the right hand and the left hand (in totally four-measure). However, only the maximum measure was considered for the study.

d) A statement as to whether your sample can be considered representative of a larger population.

Answer: The sample size was carefully calculated to ensure that the results obtained for this sample could be extrapolated to the population of workers in the automotive industry. 

e) A description of how participants were recruited, and f) descriptions of where participants were recruited and where the research took place.

Answer: The entire study took place at the assembly area facilities .In the present study, the sample was recruited from a population of 1225 direct workers, using the randomization criterion, from the entire list of employees in the assembly area, provided by the company.

5. Please ensure that you refer to Figure 4 in your text as, if accepted, production will need this reference to link the reader to the figure.

Answer: It was modified in the manuscript. 

6. Please include captions for your Supporting Information files at the end of your manuscript, and update any in-text citations to match accordingly. Please see our Supporting Information guidelines for more information: http://journals.plos.org/plosone/s/supporting-information

Answer: It was modified in the manuscript. 

2- Reviewer 1:

1. Abstract: do not repeat the words that were used in the title as key terms.

Answer: The key words were modified the manuscript:

 ”Key terms: Handgrip strength; Sarcopenia; Functional thresholds; Work population

2. Line 161: correct reference 9 (remove parentheses and inform author).

Answer: It was corrected by:

 “Furthermore, the prevalence of workers in each of the weak handgrip strength groups was determined through 1 SD and 2 SD below the peak mean values, also in relation to the low grip strength cut-off for men at 27 kg and for women at 16 kg, based on Cruz-Jentoft et al. (9).”

3. Results: The description of the sample was well done, but I missed the number of individuals per age group. I request that the authors add this information.

Answer: “ It was included, in the tables, the reference of the number of workers for each age group. Additionally tables have been improved “.

4. The results were well explored as text, however, the tables and figures can be improved.

5. Tables and figures:

Complement in the title the characteristics of the studied population: Portuguese women and men workers in the automotive industry.

Include the description of the abbreviation OR avoid abbreviation and include the complete nomenclature.

Include the number of people per group (“N”), when necessary.

Answer related to question 4 and 5: “It was changed as suggested, in the manuscript “.

6. Figures 3a and 3b: The figures are not clear; I suggest inserting the units of measurement of the scales.

Answer: “The figure was changed to figure 3 and figure 4, according the suggestions’.

3- Reviewer 2:

7. Abstract

Results: Please, verify age range for women regarding force declining. The abstract indicated force starting to declines for ages bellow the range that force reaches the peak … is it correct?

Answer: “The age interval is correct “. 

8. Conclusion: authors re-introduced the issue and suggested applications in very broad sense. Please, re-write the text according to the evidences of your results and make specific appointments concerning occupational care of Portuguese workers.

Answer: The sentence was changed:

“Normative values can help delineate the career path of workers because they portray risk values according to age, height, and gender. The normative values help health and engineering professionals and ergonomists to adjust the tasks demands to the morphological and strength characteristics of the workers “. 

9. Introduction

The present study aimed “to identify the normative values of handgrip strength for Portuguese workers”. According to the authors, the assessment of strength level would support the diagnosis of muscle syndromes related to aging process (e.g.: frailty and sarcopenia), and handgrip scores is reliable to parametrize the development of syndromes affecting muscle ability to generate force accordingly. Moreover, authors included some paragraphs (apart of general introduction, which does not match PLOS One guidelines) to specify how different countries developed their own reference for handgrip strength.

However, author did not give enough information to support the theoretical link between handgrip scores of muscle force and muscle weakness/disability to perform a given occupational function. Furthermore, author fails to state the need to develop an index of force for each country, and the reasons for the lack of confidence in worldwide scores.

Answer: It was added the following text in the introduction:

“One of the key measures used for the diagnosis of these age-related syndromes is the handgrip strength test (HGS), which is measured by the static force exerted by the hand when holding and tightening a dynamometer (13). The HGS is considered a biomarker of healthy aging (40) and an indicator of general muscle strength (14). Also, it has excellent inter-rater reliability, is easy to apply, and has a low cost (4,10-12). 

Studies are rarely carried out on active-age workers who are exposed to highly demanding activities, such as in the automotive industry.

Reference values regarding handgrip strength in the automotive industry are few and feature small samples, being n 161, in the Australian population and 206 individuals in the British population (33, 38). By contrast there are studies with larger sampling, such as the study carried out in Germany with a sample of 11,790 people aged 17-90 years, but the type of occupational activity that people were allocated to was not categorized. This Germany study validated the strong association between body height and increased handgrip strength, with each 10 cm in height associated with an increase in handgrip strength of 2 to 4 kg (24). In another study carried out in the British population, the sample was 1645 people but did not report the activities the participants performed. 

 A study developed in the American population which is widely used in research because it was one of the first to provide normative values of handgrip strength, had a sample of 638 people between 20 and 90 years of age. However this study did not categorize the type of activities and occupational factors the participants were exposed to. (27). 

Therefore, studying the HGS measure in active-age workers can be fundamental in avoiding the early appearance of age-related syndromes and even the early appearance of work-related musculoskeletal disorders, which would influence the decrease in premature retirements.”

10- Methods

10- pp. 56 – 57: why to elect only workers from final assembly area?

Answer: “It is worth mentioning that the assembly area in the automotive industry is the most critical place, regarding the high rate of musculoskeletal disorders, mainly in the upper limb “.

11- pp. 60 – 65: text discussing the sample size. It is not appropriate in method section.

Answer: “It was removed from that section and placed in the discussion part.

12- pp. 67-69: include the age range, numbers of workers, average weight and height for each age group.

Answer: “As the aim of this study was to determine the threshold of handgrip strength according the height, we still did not collect the weight data. However, the data related to height, number of employees by age group and sex, was carried out and is included in tables 1, 2, and 3 “.

13- pp. 70-74: include the numbers of participants excluded according each criteria.

Answer: We add the following text: 

“To ensure normative values for healthy workers, the following exclusion criteria were applied: a) minimum value of 10 kg of handgrip force; b) present any medical restriction and occupational disease from the industry's occupational department; c) SF-12 criteria (Short Form Health Survey) with a score lower than 5% of the physical component score of the quality of life scale (24).”

14- p. 76: Methods (again?) …. Measurements (?)

Answer: It was changed to Measurements

15- pp. 114 – 117: identify the dependent parameter.

 p. 115: OLS require alpha number to determine include and excluded variables.

pp. 120-122: support the choice of 1SD and 2SD to define cut-off values.

 pp. 124 – 125: support the choice of 1SD to define risk threshold.

Answer: The text was rewritten:

“Descriptive statistical analysis was used to determine mean (M), standard deviation (SD) and median values (ME) of the handgrip strength, stratified by five age groups of the women and the six age groups of men. Sex-specific profiles of handgrip strength were designed by the Ordinary Least Square regression (OLS) analysis (16), where height, age, age squared and height squared (independent variables) are entered in the models as determinant factors of the maximum grip strength (dependent variables) in both female and male groups. The objective of this statistical technique was to verify the mean peak values for women and men groups. A p-value of 0.05 was set as a statistically relevant result.

Afterwards, the cut-off values were calculated in order to identify workers with weak grip strength. These values were defined as 1SD and 2SD below the mean peak value and stratified by gender (17; 9). The resulting values were plotted from the age groups defined by height. 

Finally, handgrip strength measurements were first standardized for age and body height. The risk threshold was determined by subtracting 1 SD from the mean value of height-adjusted handgrip strength and age group (9). The standardized measures of handgrip strength values were the z-standardized residuals (derived from sex-specific OLS regressions of handgrip strength values in kg on age (in years) and body height (in cm). 

16- Results:

pp. 131 – 140: describe the process of sample selection and should be located accordingly. Decimal cases should be separated by points.

Answer: The paragraph was changed: 

“A cross-sectional study was conducted in the area of assembly of the automotive industry. In the present study, the sample was recruited from a population of 1000 direct workers, using the randomization criterion, from the entire list of employees in the assembly area, provided by the industry's occupational area. The entire study took place at the assembly area facilities, lasting 20 weeks from September 2018 to January 2019.”

17-pp. 141-144: I can’t observe differences in force values when compared the age groups for both female (Fig 1) and male (Fig 2). Did authors proceeded statistical test to compare force between groups?

All Figures should be revised. The units are not present, as well as, “Y” title.

Answer: “There was no association between the groups of women and men. However it was stablished that the determination of normative values, it is more appropriate to perform it by gender. The figures have been revised and changed “.

18- p. 150: I am not sure that the term “decline” fits well to describe force profile variations between age groups. If authors considered such score the worsts, then force has not age as independent factor among females.

Answer: The sentence was rewritten:

“The most pronounced HGF peak decrease appeared in women in the age group of 30-34 year-olds, with 4kg (Fig 1) below the mean peak strength.2

19- p. 163: revise kilograms symbol; Figures 3a and 3b aren’t parts of the same figure.

Answer: “The figures were transformed to 3 and 4, and the kilogram symbol was also changed “.

20- e) Discussion

Question 16: pp. 231-233: the findings are limited to the establishment of reference values. There are no results to confirm that workers having force values bellow the risk threshold presented sarcopenia or frailty syndrome. Please, revised this statement or support it better from your results.

Answer: The sentence was rewritten: 

“The study established normative values for the handgrip strength of Portuguese workers in the automotive industry to identify low handgrip strength limits. Such limits can assist the trigger in identifying, declining aspects of muscle capacity, and even an alert to check for the possible development of frailty and sarcopenia syndrome. These declines in muscle capacity and the warning for syndromes related to aging are likely since the grip strength measure can be used as a predictor of whole-body strength (42). As for fragility and sarcopenia, the HGS is one of the parameters to identify them (9).”

21- pp. 236-237: This statement could not be corroborated from the results of Tables 1 and 2. Please, discuss it better.

Answer: The sentence was improved

“However, in the present study, when comparing the lowest height and the highest height, in both groups (table 1 and table 2), it is interesting to note that sometimes the difference is greater than 2 to 4 kg of strength (24). This may also be related to the type of activity that workers perform in the assembly area (21), where the muscle groups referring to the forearm region, maybe more exercise (1). It is important to note that the study population is healthy workers.”

22- Question 17: p. 272: “male population” means “male population”?

Answer: It was changed as suggested by the male population.

23- Question 18: pp. 289 – 300: results for peak of force, force decrement, and cut-off points were just comment and not compared to others results from literature.

In discussion section, authors limited to explain differences regarding the references of force for handgrip test by comparing with others similar studies, or even reinforce main results. The applications to parametrize muscle syndromes and support occupational adjustments were discussed superficially.

Answer: “Considering the scarcity of studies that establish normative values and the fact that sarcopenia and frailty have not been evaluated, we consider it too relevant to conduct the discussion in the light of studies that had the same objectives “. 

24-Conclusion

Question 19 : pp. 304-309: author concluded about possible applications of the results. Most of these applications had no results to support the appointments. For example: (1) the ability to perform work above head was not related to handgrip force, and (2) the clinical relevance of handgrip force ability or disability to endorse worker relocation should be the proposition for future studies.

Answer: The sentence was rewritten and simplified

“Normative values can help delineate the career path of workers because they represent risk values according to age, height, and gender. These thresholds can be very useful to help on the adjustment of work conditions to the morphological and strength characteristics of the worker. Thus, it is possible to design or redesign the conditions work processes associated with the predictive values of HGS and the implementation of the workers' clinical surveillance system through periodic using the HGS test.”

---

## [Decision Letter · Decision Letter 1]

10 Jul 2020

Normative reference values of the handgrip strength for the Portuguese workers.

PONE-D-20-06221R1

Dear Dra. Sarah Moreira Fernandes Bernardes

We’re pleased to inform you that your manuscript has been judged scientifically suitable for publication and will be formally accepted for publication once it meets all outstanding technical requirements.

Kind regards,

Anderson Saranz Zago, PhD

Academic Editor

PLOS ONE

Reviewers' comments:

Reviewer's Responses to Questions

**Comments to the Author**

1. If the authors have adequately addressed your comments raised in a previous round of review and you feel that this manuscript is now acceptable for publication, you may indicate that here to bypass the “Comments to the Author” section, enter your conflict of interest statement in the “Confidential to Editor” section, and submit your "Accept" recommendation.

Reviewer #1: All comments have been addressed

Reviewer #2: All comments have been addressed

2. Is the manuscript technically sound, and do the data support the conclusions?

Reviewer #1: (No Response)

Reviewer #2: Yes

3. Has the statistical analysis been performed appropriately and rigorously? 

Reviewer #1: (No Response)

Reviewer #2: Yes

4. Have the authors made all data underlying the findings in their manuscript fully available?

Reviewer #1: (No Response)

Reviewer #2: Yes

5. Is the manuscript presented in an intelligible fashion and written in standard English?

Reviewer #1: (No Response)

Reviewer #2: Yes

6. Review Comments to the Author

Reviewer #1: (No Response)

Reviewer #2: Authors addressed all my previous comments and improved text, results and overall soundness of the manuscript.

7. PLOS authors have the option to publish the peer review history of their article (what does this mean?). If published, this will include your full peer review and any attached files.

Reviewer #1: No

Reviewer #2: **Yes: **Dalton Müller Pessôa Filho

---

## [Editor Report · Acceptance letter]

17 Jul 2020

PONE-D-20-06221R1 

Normative reference values of the handgrip strength for the Portuguese workers. 

Dear Dr. Bernardes:

I'm pleased to inform you that your manuscript has been deemed suitable for publication in PLOS ONE. Congratulations! Your manuscript is now with our production department. 

Kind regards, 

on behalf of

Dr. Anderson Saranz Zago 

Academic Editor

PLOS ONE